# Image Degradation Model for Dynamic Star Maps in Multiple Scenarios

**Haima Yang** [1,2]**, Yan Jin** [1]**, Yinan Hu** [3,]*****, Dawei Zhang** [1]**, Yong Yu** [4]**, Jin Liu** [5]**, Jun Li** [1]**, Xiaohui Jiang** [6] **and Xiaojun Yu** [7]

1 School of Optical-Electrical and Computer Engineering, University of Shanghai for Science and Technology, Shanghai 200093, China
2 Key Laboratory of Space Active Opto-Electronics Technology, Chinese Academy of Sciences, Shanghai 200083, China
3 Shanghai Institute of Technical Physics, Chinese Academy of Sciences, Shanghai 200083, China
4 Shanghai Astronomical Observatory, Chinese Academy of Sciences, Shanghai 200030, China
5 School of Electronic and Electrical Engineering, Shanghai University of Engineering Science, Shanghai 201620, China
6 School of Mechanical Engineering, University of Shanghai for Science and Technology, Shanghai 200093, China
7 Shanghai East China Railway Electrification Engineering Co., Ltd., Shanghai 200041, China
***** Correspondence: 12120082@bjtu.edu.cn

**Abstract:** To meet the ground test requirements of star sensors, we establish the star map simulation algorithm and the interactive interface in multiple scenarios. The combination of the degradation model of star points, the imaging noise model, and the attitude disturbance model is introduced to solve the problem of different patterns of noise existing in the actual measurement, improving the traditional simulation model. In addition, a user-friendly interface design makes it easier for both scholars and average individuals to understand the parameters and then generate static single-frame star maps—or a series of dynamic sequence star maps—under various conditions. The results of the proposed star map simulation method are highly comparable to the actual captured star images, and this method can be applied for the tests and calibrations of star sensors.

**Keywords:** star map; image degradation; dynamic environment; aerospace; noise model

## 1. Introduction

Star sensors play an essential role in aerospace attitude detection. Star sensors need to undergo rigorous performance tests before the engineering of applications. Generally, there are three ways to calibrate star sensors with high accuracy: numerical simulation, hardware simulation, and outfield observation experiments. Outfield observation experiments are conducted by placing a star sensor in an open area with low atmospheric turbulence and relatively weak stray light. Such an approach is costly and time-consuming and is usually only used as the last step in a star sensor validation experiment. Hardware simulation refers to utilizing a starfield simulator in a laboratory environment to simulate an infinity star point. The star sensor carries out identification and attitude calculation by observing the simulated star images. The disadvantages of this method are that the starfield simulator is expensive, and the parameters of the starfield simulator, such as field of view and aperture—which are difficult to match precisely with the star sensor—are not adjustable. Therefore, the versatility of the hardware simulation is limited. Compared with these two methods, numerical simulation is a highly flexible and the least resource-intensive approach. In routine performance tuning or laboratory experiments, simulated star maps are widely used because of their low cost and ease of use [1–4].

Many scholars have invested in studying the use of star sensors from digital simulation, i.e., the study of star map simulation algorithms. In a star map simulation, the

establishment of the noise model is essential. To solve this problem, some scholars simulated the Gaussian grayscale distribution of star point images by considering the stellar image shift caused by the satellite motion [5]. Some scholars have introduced new theories, such as the effect of rotation around the optical axis on star point imaging, the star color index, and the idea of black-body radiation, to evaluate the quality of star points in a simulation result [6,7]. To enlarge the simulated scenarios in the simulation system, nebulae, moonlight, and Earth-obscured background are added into the field of view in the star sensors [8,9]. The literature [10] proposed a fast pixel-discretization algorithm based on the convolutional surface model to simulate dynamic real-time star maps and established a complete description of the star point motion trajectory model. However, the parameters in this algorithm are difficult to understand, limiting its usage. Researchers [11] designed a star map simulation algorithm for arbitrary exposure time length in a high-fidelity manner. However, this method is performed in an ideal state, without incorporating various error factors into the model.

Here, we design a set of algorithms and interfaces for simulating star maps in multiple scenarios from a numerical simulation approach. The algorithm includes the simulation of static single-frame star maps and dynamic sequence star maps, taking into account various noise sources. The noise may come from the electronic noise caused by the imaging system, the overall noise amplitude variation due to atmospheric condition or stray light, and the attitude disturbance caused by the different motion states between the imaging platform and the tracking star target. With various parameter inputs, noise levels and dynamic trajectories are visualized, and star maps close to the actual images are generated, which can provide a source of data for the parametric testing of the star sensor.

## 2. Motion Blur Degradation Model

### 2.1. The Process of Image Degradation

For a ground-based star sensor, during the process of image acquisition, there will be many challenges, such as the shift and superposition of light beams. All these difficulties can result in the degradation of image quality [12]. Figure 1 shows the general model of image degradation.

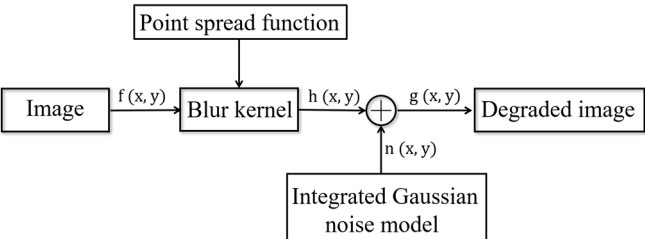

**Figure 1.** The general model of image degradation.

In Figure 1, if h (x, y) is a linear spatially invariant process, then the degraded image in the spatial domain is represented as:

$$g\,(x, y) = h\,(x, y) \times f\,(x, y) + n\,(x, y) \tag{1}$$

in which f (x, y) represents a static, two-dimensional image, which degenerates to g (x, y) under the interference of additive noise n (x, y) through blur kernel h (x, y). * denotes a general convolution operator.

### 2.2. Simulations of Defocus Factor

In general, h (x, y) in the degradation model includes defocus blur and motion blur. To realize the out-of-focus impact more accurately, we use a point spread function model to simulate the image degradation with the following formula [13,14]:

$$g_{ij} = \frac{A}{2\pi\sigma^2} \int\limits_{i-0.5}^{i+0.5}\int\limits_{j-0.5}^{j+0.5} exp[-\frac{\left(x_{i,j} - x_m\right)^2 + \left(y_{i,j} - y_m\right)^2}{2\sigma^2}]dxdy \tag{2}$$

where A denotes the energy grayscale coefficient, which is related to the total illumination, i.e., magnitude, of the imaging point on the photosensitive surface of the star sensor. $\sigma$ denotes the Gaussian dispersion radius of the energy distribution in the star point region, indicating the degree of defocusing. $(x_{i,j}, y_{i,j})$ is any pixel within the range of the diffuse pixel point. $(x_m, y_m)$ denotes the projected position of the star on the imaging plane of the star sensor. We can evaluate a static star point in a simulation in terms of two key parameters: the gray energy factor A and the Gaussian dispersion radius $\sigma$ [15]. Table 1 lists the parameters of the simulated image in Figure 2, which shows the combination of different values of A and $\sigma$, representing various results of image degradation that may occur under real circumstances.

**Table 1.** Parameters of the simulated image.

| Position | A | $\sigma_x$ | $\sigma_y$ |
|----------|-----|-----|-----|
| (15, 15) | 613 | 1.0 | 1.0 |
| (30, 30) | 2113 | 2.0 | 1.5 |
| (40, 10) | 4113 | 2.5 | 3.0 |

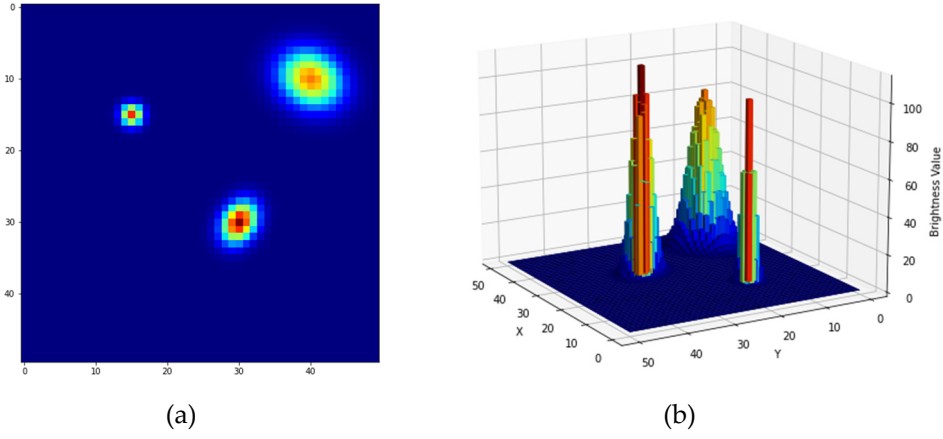

|     |     |
|-----|-----|
| (a) | (b) |

**Figure 2.** The combination of different A and $\sigma$: (**a**) 2D; (**b**) 3D.

### 2.3. Integrated Gaussian Noise Model

In addition to the simulation for the background star point and target star point, the star sensor simulation also needs to simulate the effect of noise to improve the simulation's authenticity. The random noise contained in the star map is mainly ambient noise and detector noise, such as dark current noise and output noise. These types of noise can be defined as Gaussian noise, due to their randomness [16]. The literature [2,7] states that the simulation of ground-based imaging systems must consider the effects of atmospheric influences and uneven background illumination, such stray light, moonlight, and sunlight, on the captured images. These effects will lead to a change in the overall brightness value

of the image from weak to strong. Here, we use Equation (3) to represent the integrated Gaussian noise model:

$$N_{img} = f(x, y) \times \psi \times \eta(0, std), \eta \sim \text{Gaussian}(0, std) \qquad (3)$$

where $f(x, y)$ is the original image. $\psi$ is designed to control the shape of the noise area, and it can be expressed by functions. Here is an example of a linear expression. If we set $\psi_y$ $(w, k) = w \times y + k$, a mathematical model of monotonic increase or monotonic decrease, the image result will have a gradient effect on the y-axis. That is, the magnitude of the image will change from dark to light or from light to dark. $\psi$ can also be expressed in terms of other functions, such as polar coordinates. With the use of polar coordinates, a circular, radial noise pattern can be clearly expressed. $\eta(0, std)$ is the amount that controls the magnitude of the noise, which follows a Gaussian distribution. The default mean of $\eta$ is 0, and std represents the standard deviation. The larger the value of std, the larger the range of noise fluctuation.

Figure 3 shows the simulations of the integrated Gaussian noise model with different inputs. The simulated image is 8 bits, and the image array is 512 × 512, where the Gaussian dispersion radius of the star point is between 1 and 4. The star point circled in the figures is the same star point (with the same A, σ, and location) in the three images.

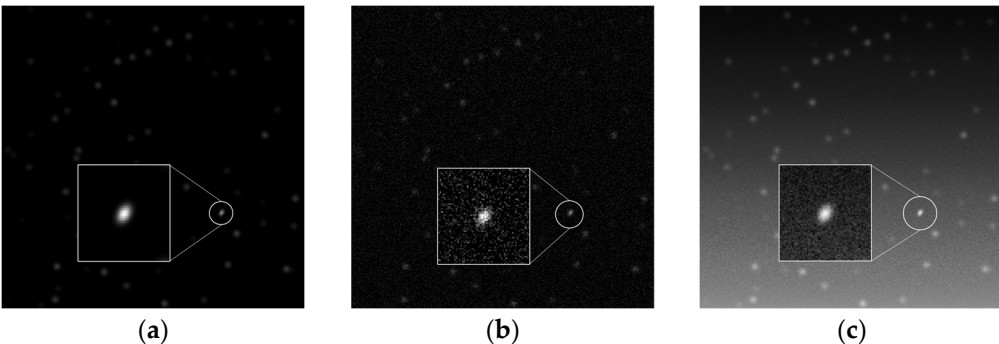

| (a) | (b) | (c) |

**Figure 3.** The simulations of the integrated Gaussian noise model with different inputs: (**a**) the simulation result without the noise model; (**b**) the simulation result with $N_{img} = f(x, y) \times 1 \times \eta(20, 5)$; (**c**) the simulation result with $N_{img} = f(x, y) \times \psi_y \cdot \eta(20, 5)$, $\psi_y = y + 5$.

## 3. Simulations for Dynamic Sequence Star Maps

### 3.1. The Energy Distribution Model of Trajectory

Because of the relative motion between the ground-based imaging system and the celestial body, this study will complete a star map simulation in a dynamic environment. In the dynamic environment, the target star point will be dragged to form a trajectory on the imaging plane during the camera exposure time, which is called a "smearing star point." The length and direction of this trailing path can reflect the motion information of the target star point [17]. To summarize, if there is a relative motion between the star point and the ground-based imaging system, the target is in the form of a trajectory line segment. Particularly if the imaging system has tracked and located a target object, the target star point appears relatively stationary in the image.

To achieve a star streak quickly, the exposure time T is divided into N segments at equal intervals. Accordingly, the trailing path of the star point is also divided into N segments. When interval $\delta = T/N$ is short enough, each part of the trailing path can be approximated as a point. Through such an approximation process, the dynamic trajectory of the star point can be approximated as a superposition of N static points. We set the total energy during the exposure time of the star point to $E_0 = \sum_{n=1}^{n=N} E_{i,i+1}$, and the energy distribution of each

static star point simulating the dynamic trajectory is shown in Equation (4). In the interval from t = i to t = i + 1, the simulation formula for the energy of a star point is:

$$E_i(x, y) = \frac{E_{i,i+1}}{2\pi\sigma^2} \int \int \exp[-\frac{(x - x_i)^2 + (y - y_i)^2}{2\sigma^2}]dxdy \tag{4}$$

Assuming that the dispersed spots of the above Gaussian distribution are equal in radius, the energy distribution function of the star image in each small exposure time is superimposed, and the total energy distribution function of the star image is as follows:

$$E(x, y) = \sum_{n=1}^{n=N} E_i(x, y) \tag{5}$$

*3.2. Attitude Disturbance Model*

When the ground-based imaging system tracks a moving target star point, there are attitude disturbances between the measurement. The attitude disturbance mainly refers to the jitter of the ground-based platform brought about by its working condition and the motion lag of the camera system when it does not accurately track the motion of the target object.

Figure 4 shows a schematic diagram of the projection trajectory of the projection of the star point on the imaging plane under the influence of attitude disturbance. When rotating in any direction within $O_s x_s y_s z_s$, the coordinate system measured around the star sensor, the angular velocity can be decomposed onto the 3 axes of $O_s x_s y_s z_s$. By setting the star image on the imaging plane Oy on the fixed axis of the uniform angular velocity motion along the clockwise direction, the angular velocity is recorded as $\omega_z$. By setting the ith instant in time, the star point position vector $R_i = (x_i, y_i)$, $|R_i| = \sqrt{x_i^2 + y_i^2}$. The angle between the $R_i$ and the Oy axis is denoted $\theta_i$, so the velocity of the star at the ith instant in time can be expressed as:

$$\begin{cases} v_{ix} = \omega_z|R_i|\cos\theta_i + v_{0x} = \omega_z y_i + v_{0x} \\ v_{iy} = -\omega_z|R_i|\sin\theta_i + v_{0y} = -\omega_z x_i + v_{0y} \end{cases} \tag{6}$$

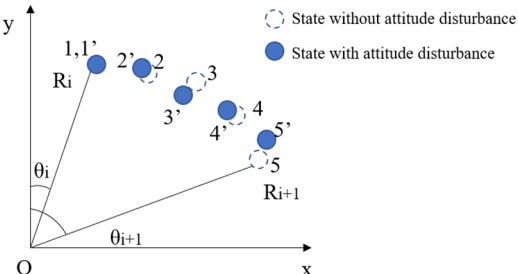

**Figure 4.** The schematic of attitude disturbance.

Given the time elapsed from the ith instant in time to the i + 1st instant in time, the amount of star point displacement [18] can be expressed as:

$$\begin{cases} \Delta x_i = x_{i+1} - x_i = v_{ix} \cdot \Delta t = (\omega_z y_i + v_{0x}) \cdot \Delta t \\ \Delta y_i i = y_{i+1} - y_i = v_{iy} \cdot \Delta t = (-\omega_z x_i + v_{0y}) \cdot \Delta t \end{cases} \tag{7}$$

To achieve the effect of motion blur, we propose that $N_{\Delta x}$ be added as a blur factor to the expression of the appropriate amount of the star position:

$$\begin{cases} x_{i+1} = x_i + \Delta x_i + N_{\Delta x}, N_{\Delta x} \sim \text{Gaussian}(0, n) \\ y_{i+1} = y_i + \Delta y_i + N_{\Delta y}, N_{\Delta y} \sim \text{Gaussian}(0, m) \end{cases} \tag{8}$$

where the default mean of the Gaussian distributed random number is 0. n and m represent the standard deviation of the set Gaussian random number. The values of n and m can be same. The larger the n and m, the greater the deviation of the simulated star point from the preset position.

In conclusion, the description of the star trajectory can be expressed by Equation (9):

$$\begin{cases} x_i(t) = x_{i0} + \int_{t0}^{t0+T} v_{ix}(\tau)d\tau \\ y_i(t) = y_{i0} + \int_{t0}^{t0+T} v_{iy}(\tau)d\tau \end{cases} \tag{9}$$

Therefore, we use the proposed methods to simulate the dynamic trajectory of a star point. Figures 5–7 show the superposition of the different states when the simulated star trajectory is a linear equation according to $\begin{cases} x_{i+1} = x_i + 1 + N_{\Delta x} \\ y_{i+1} = y_i + 1 + N_{\Delta y} \end{cases}$. Figure 8 shows the superposition of the different states when the simulated star trajectory is a curvilinear equation according to $\begin{cases} x_{i+1} = x_i + 1 + N_{\Delta x} \\ y_{i+1} = -0.006x_i^2 + 2.7x_i - 27 + N_{\Delta y} \end{cases}$.

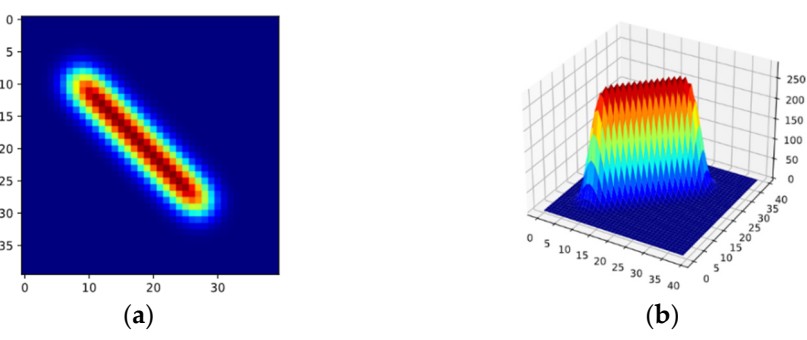

**Figure 5.** Simulation results with $A = 2000$, $\sigma = 2$, $N_{img} = f(x, y) \times 1 \times \eta(0, 0)$, $N_{\Delta x}, N_{\Delta y} \sim Gaussian(0, 0)$: (**a**) 2D; (**b**) 3D.

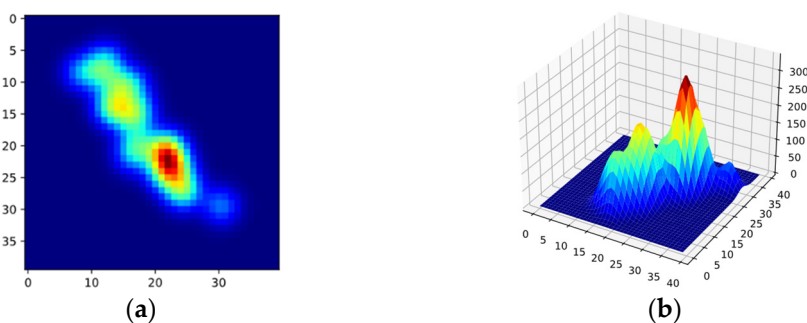

**Figure 6.** Simulation results with $A = 2000$, $\sigma = 2$, $N_{img} = f(x, y) \times 1 \times \eta(0, 0)$, $N_{\Delta x}, N_{\Delta y} \sim Gaussian(0, 2)$: (**a**) 2D; (**b**) 3D.

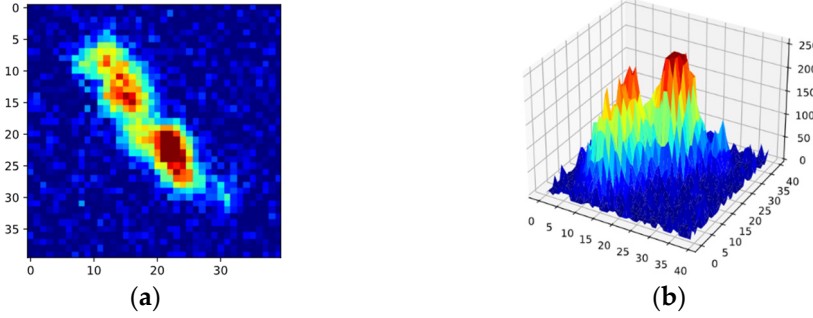

**Figure 7.** Simulation results with $A = 2000$, $\sigma = 2$, $N_{img} = f(x, y) \times 1 \times \eta(0, 20)$, $N_{\Delta x}, N_{\Delta y} \sim Gaussian(0, 2)$: (**a**) 2D; (**b**) 3D.

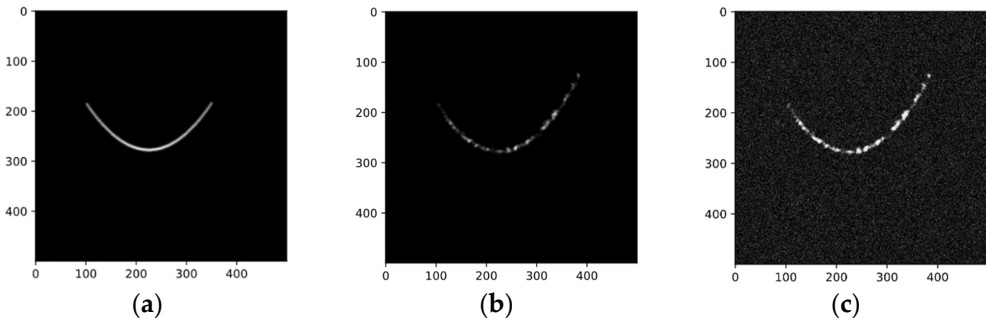

**Figure 8.** Simulation results of simulating star point trajectories as curvilinear equations: (**a**) results with A = 2000, σ = 2, $N_{img}$ = f(x, y) × 1 × η(0,0), $N_{\Delta x}$, $N_{\Delta y}$ ~ Gaussian(0,0); (**b**) results with A = 2000, σ = 2, $N_{img}$ = f(x, y) × 1 × η(0,0), $N_{\Delta x}$, $N_{\Delta y}$ ~ Gaussian(0,2); (**c**) results with A = 2000, σ = 2, $N_{img}$ = f(x, y) × 1 × η(0,40), $N_{\Delta x}$, $N_{\Delta y}$ ~ Gaussian(0,2).

## 4. Interface Function Description

The field of view of the simulated detector in the proposed method is 12° × 12°, and the size of its plane array is 512 pixels × 512 pixels. The magnitude sensitivity is 8.0 apparent magnitude, the simulated imaging system has a frame rate of 50 Hz, and the exposure time for each frame is 20 ms. The interface uses the Tkinter module based on Python 3.9.

There are two functions in the designed interface. First, it is the static single-frame star map simulation. Figure 9 shows the initial interface of this function. In this state, the default number of the target star point is 1. The adjustable parameters are the Gaussian noise level of the image, the number of background stars, the maximum value of target brightness, and the minimum value of target brightness. Users can generate single-frame simulated star maps, with specified parameters, in this mode. Additionally, a function to display images is added, which can also be used separately as a picture viewer. After generating the simulated star maps, the primary information of the target star point, such as brightness information, position coordinates, and Gaussian dispersion radius, can be viewed in "Target Star Information."

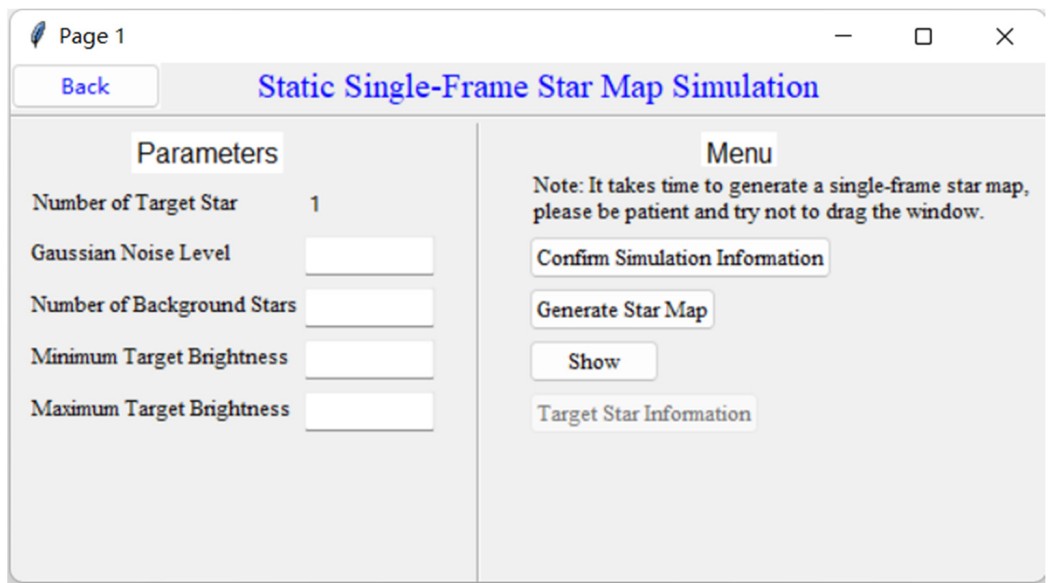

**Figure 9.** The interface of static single-frame star map simulation.

The second function is the dynamic sequence star map simulation, and its user interface is shown in Figure 10. In this mode, the simulation can provide two motion states for a target star point. One of the states is that the ground-based camera system has followed

the target point to be measured. Under this state, by default, the target in the first frame is in the center of the image. In addition to the size and brightness of a star point, it is also possible to simulate errors in the actual situation by inputting the magnitude of the attitude disturbance of a target point and the average and standard deviation of Gaussian noise. The default exposure time of each frame of the system is 20 ms. By adjusting the size of the "Dynamic Frame Count," users can input the imaging time of the simulated ground-based camera system. The other state is that the ground-based camera system has not followed the target point to be measured. Under this state, both the target and background star points have different movement direction speed sizes, compared with the ground-based camera system. The user must input the x-direction and y-direction offset to complete the exact movement simulation of the target star point. Since it would take time to generate a large number of background star points in real-time to meet the requirements, the way to handle the background star points is by loading a pre-generated binary file. This file is the sum of a specified number of star trajectories laid out according to a preset angle. The user can also change this, according to specific requirements.

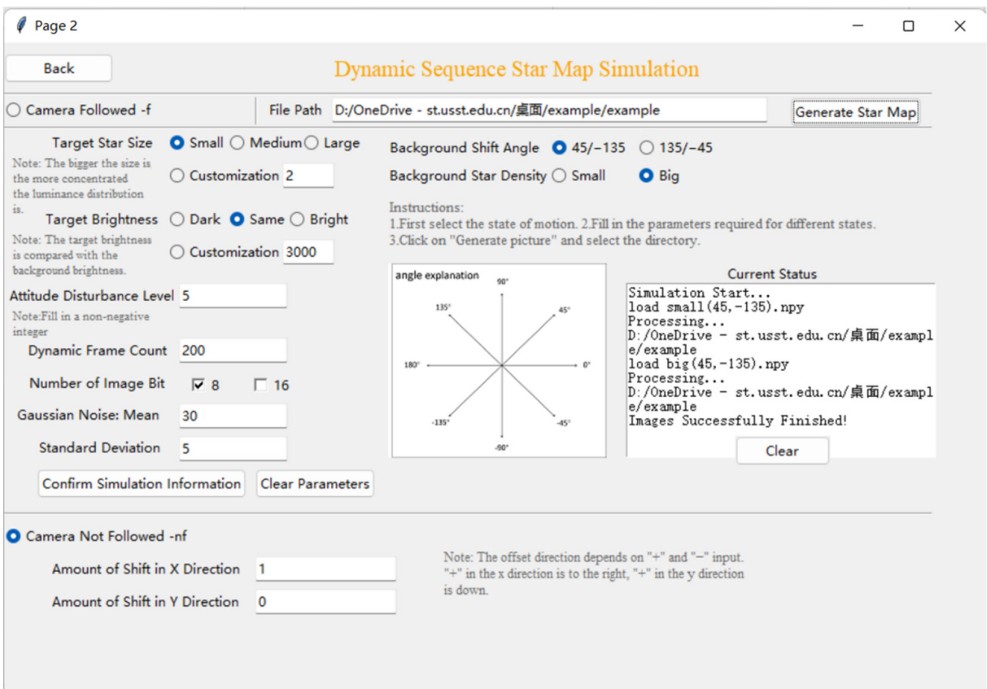

**Figure 10.** The interface of the dynamic sequence star map simulation.

Figure 11 shows the first, the 90th, and the 150th frames of the 200-frame sequence of star maps simulated under the conditions of the selected states and parameters depicted in Figure 10. The target to be measured is marked with a red circle, and the coordinates of the target star point in different frames are listed in the white textboxes.

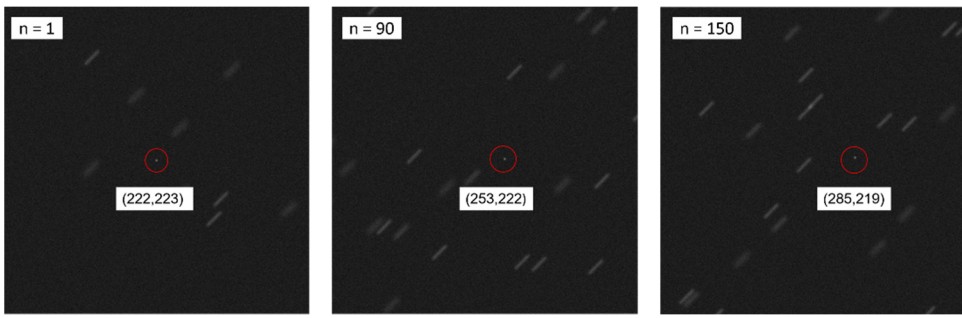

**Figure 11.** Dynamic sequence star maps.

## 5. Results

### 5.1. Image Quality Assessment

Figure 12 shows simulation results using the method proposed in this paper. Figure 12a,b shows the static single-frame star map simulation results, and Figure 12c shows the dynamic sequence star map. Figure 13 displays actual photos from the Shanghai Sheshan Observatory and from astronomer Michael A. Earl. From the gray histograms of the two types of images, we can determine that the actual photos have a broader gray distribution than that of the simulation results. Furthermore, there are some stripes in the photos. These stripes may come from the telescope lens.

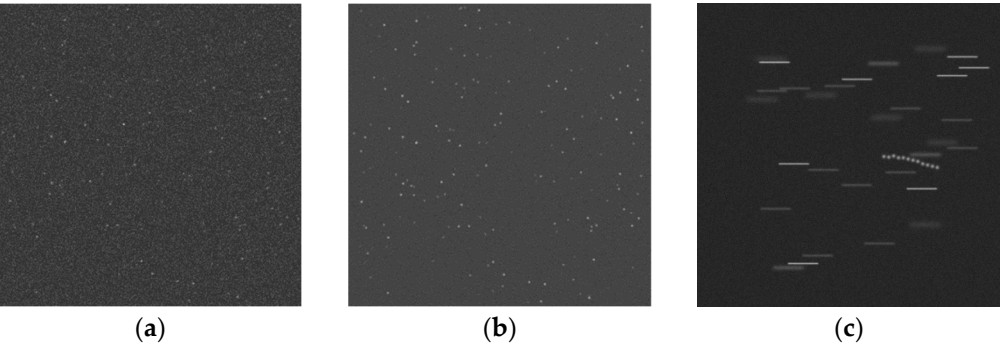

(a)                               (b)                               (c)

**Figure 12.** Simulation results using the proposed method: (**a**) simulation result for Figure 13a; (**b**) simulation result for Figure 13b; (**c**) simulation result for Figure 13c.

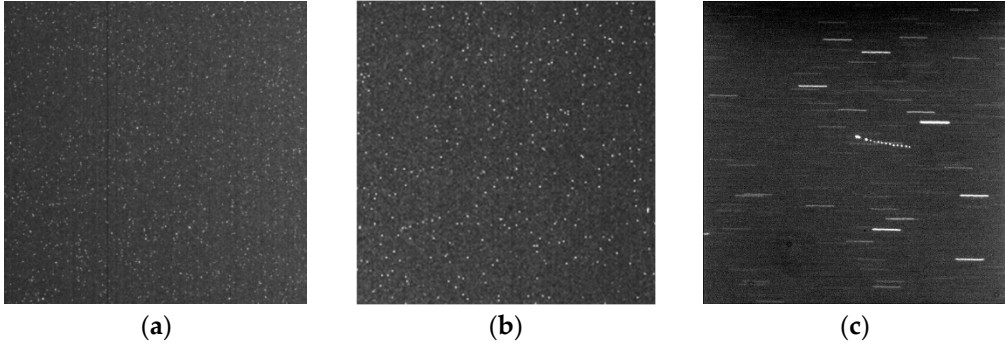

(a)                               (b)                               (c)

**Figure 13.** Real star map images: (**a**) taken by the Shanghai Sheshan Observatory; (**b**) taken by the Shanghai Sheshan Observatory; (**c**) taken by Canadian astronomer Michael A. Earl with a ground-based telescope ("Geostationary Satellite MSAT in Motion," http://www.castor2.ca/14_Images/Satellites/index.html accessed on 15 November 2021).

Here, we use two methods to evaluate the image similarity between the simulation results and actual photos. Figure 14 demonstrates the gray histograms of the simulated images in Figure 12. And Figure 15 demonstrates the gray histograms of the real images in Figure 13. First, the Bhattacharyya coefficient which is based on the gray histogram is used to indicate the color distribution of an image. The coefficient ranges from 0 to 1. The closer to 1, the more similar the two images are proved to be in terms of color distribution. The second standard is the cosine similarity, which changes an image into vectors by adding the divided gray level area numbers [19]. The calculation results are shown in Table 2, which point out that there is still a gap between the grayscale distribution of the simulated results and the actual photos, but the overall character of both is very close.

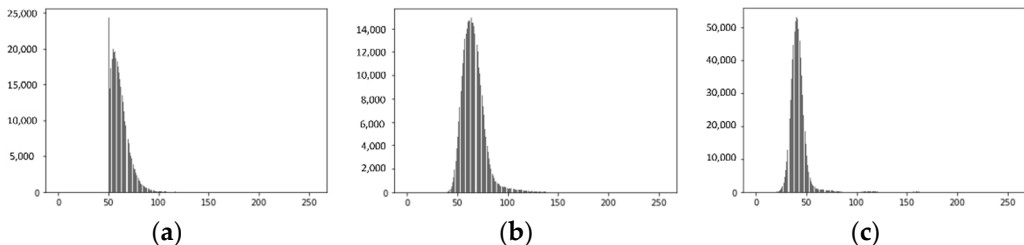

**Figure 14.** Gray histograms of the simulated images: (**a**) gray histogram of Figure 12a; (**b**) gray histogram of Figure 12b; (**c**) gray histogram of Figure 12c.

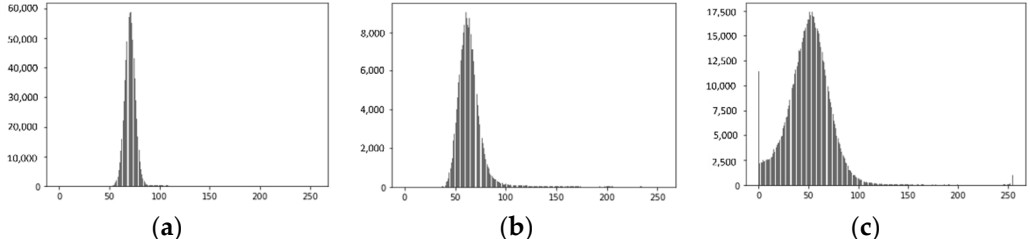

**Figure 15.** Gray histograms of the real images: (**a**) gray histogram of Figure 13a; (**b**) gray histogram of Figure 13b; (**c**) gray histogram of Figure 13c.

**Table 2.** Calculation results of image similarity.

| Number | Bhattacharyya Coefficient in Gray Histogram | Cosine Similarity |
|---|---|---|
| Figure 12a vs. Figure 13a | 0.778 | 0.977 |
| Figure 12b vs. Figure 13b | 0.639 | 0.992 |
| Figure 12c vs. Figure 13c | 0.639 | 0.893 |

Figure 12c shows the effect of a 2 s imaging time in a dynamic environment simulated using the proposed method's noise parameter $N_{img} = f(x, y) \cdot 1 \cdot \eta(0, 15), N_{\Delta x}, N_{\Delta y}, \sim Gaussian(0, 3)$. Figure 13c shows an actual image taken by Canadian astronomer Michael A. Earl with a ground-based telescope. These are some of the differences between Figures 12c and 13c and the main reasons behind them. Besides the direction of the background trajectories, which depends on the movement states of the simulated conditions, the trajectory style of the target is more noticeable in the actual photos. Because the satellite tumbles at a regular interval within an extended period, the satellite's orbit produces a periodic flickering phenomenon, which is not addressed by the method in this paper. Next, the noise distribution in Figure 13c is more complicated than in Figure 12c. Figure 13c becomes brighter due to the height of the image, which means the average of Gaussian noise becomes bigger. which means the average of Gaussian noise becomes bigger. In addition, some black dots and black areas exist in the figure in a more obvious way. It is necessary to determine different characteristics and modes of noise to improve the quality of the simulation results.

### 5.2. Star Centroid Extraction

The centroid of stars is the most important factor in attitude determination. We adopt the common centroid method to evaluate the accuracy of the simulation result of the proposed method. Figure 16a shows an actual image taken by Michael A. Earl. Four distinct targets are circled in the image, and the simulation result is shown in Figure 16b.

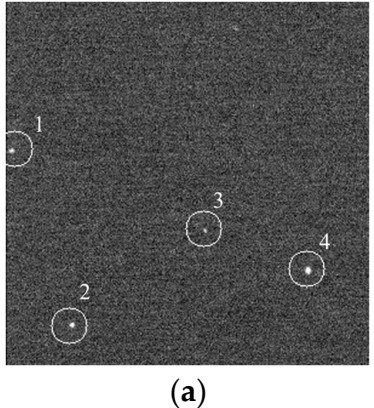
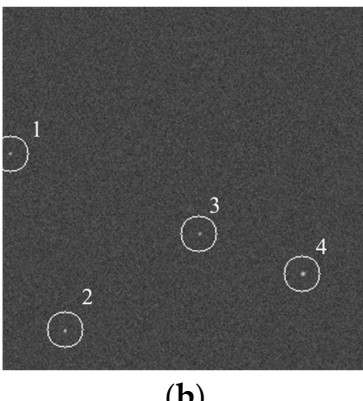

<div style="text-align:center">(**a**)        (**b**)</div>

**Figure 16.** Star centroid extraction comparison: (**a**) courtesy of Mike Earl—CASTOR; (**b**) simulation result of the proposed method.

Next, before we perform the star centroid extraction process, we need to implement the Gaussian filter to reduce the interference from background noise in Figure 16a. The Gaussian kernel size is $5 \times 5$ and the Gaussian kernel standard deviation is computed with the formula provided by OpenCV (getGaussianKernel(), https://docs.opencv.org/4.0.0/d4/d86/group__imgproc__filter.html#gac05a120c1ae92a6060dd0db190a61afa accessed on 15 November 2021). Finally, we adopt the common centroid method to extract the "center of mass" as follows:

$$\begin{cases} x_0 = \dfrac{\sum_{i=1}^{m} \sum_{j=1}^{n} x_i f_{ij}}{\sum_{i=1}^{m} \sum_{j=1}^{n} f_{ij}} \\ y_0 = \dfrac{\sum_{i=1}^{m} \sum_{j=1}^{n} y_j f_{ij}}{\sum_{i=1}^{m} \sum_{j=1}^{n} f_{ij}} \end{cases} \tag{10}$$

where $f_{ij}$ denotes the value of the image after background noise reduction.

The same steps will apply to the detection of Figure 16b. The results for the two images are shown in Table 3.

**Table 3.** Results of star centroid extraction in Figure 16a,b.

| No. | Figure 16a | | Figure 16b | | Error | |
|---|---|---|---|---|---|---|
| | **x** | **y** | **x** | **y** | **Δx** | **Δy** |
| 1. | 7.329 | 121.973 | 7.498 | 121.983 | 0.169 | 0.01 |
| 2. | 52.113 | 267.078 | 52.037 | 266.969 | −0.076 | −0.109 |
| 3. | 163.037 | 187.762 | 162.989 | 187.964 | −0.048 | 0.202 |
| 4. | 248.171 | 220.978 | 247.978 | 220.970 | −0.193 | −0.008 |

Where the errors are calculated as the follow equation:

$$\begin{cases} \Delta x = x_\beta - x_\alpha \\ \Delta y = y_\beta - y_\alpha \end{cases} \tag{11}$$

where $x_\beta$ and $y_\beta$ denote the x and y coordinate values of the stars in Figure 16b. $x_\alpha$ and $y_\alpha$ denote the x and y coordinate values of the stars in Figure 16a.

As we can see, because of the noise of figures, the accuracy is not perfect, but it is acceptable for common use. If we perform the image processing in detail and apply a more specific centroid extraction method, the error will be smaller.

## 6. Conclusions

In summary, this paper carries out a simulation method for dynamic sequential star maps of space targets in multiple scenarios with complex star backgrounds. Numerical simulation provides a more convenient way to generate the abundant data of star maps to promote the study of star sensors. A static star point model and a dynamic sequence star map model are constructed, respectively. We take the random noises, such as detector noise and dark current noise, and uneven environment illumination into account. Meanwhile, the impact of the noise model analysis and platform motion disturbances on imaging will also provide a reference for performance testing and verification of star sensors. In the next stage of the work, we may consider the condition in which multiple interference target star points exist. In addition, it is essential to continue to study the noise sources and patterns in authentic images so that the parameters can be further optimized, making them more similar to those in the actual scenario.

**Author Contributions:** Conceptualization, H.Y.; methodology, Y.H.; software, Y.J.; validation, Y.H. and D.Z.; formal analysis, Y.Y. and J.L. (Jin Liu); investigation, J.L. (Jun Li); writing—original draft preparation, Y.J.; writing—review and editing, Y.J. and H.Y.; project administration, X.J. and X.Y. All authors have read and agreed to the published version of the manuscript.

**Funding:** This work was supported in part by the Joint Astronomical Fund of the National Natural Science Foundation of China (grant number U1831133), the Key Laboratory of Space Active Opto-electronics Technology of the Chinese Academy of Sciences (grant number 2021ZDKF4), and the Shanghai Science and Technology Innovation Action Plan (grant numbers 21S31904200,22S31903700).

**Institutional Review Board Statement:** Not applicable.

**Informed Consent Statement:** Not applicable.

**Conflicts of Interest:** The authors declare no conflict of interest.

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
