# Peer review of "Image Degradation Model for Dynamic Star Maps in Multiple Scenarios"

_photonics, doi:10.3390/photonics9100673_

Round 1

Reviewer 1 Report

Page 2, Figure 1, rightmost box:
Where is written "Degradation image" should read "Degraded image".

Page 3, Equation 2 and line 89:
It would be less confusing if the pixel (x,y) were denoted by (i,j).
Hence g_ij(x,y) in Equation 2 would become g_ij.

Page 3, line 109: In "where f (x, y) is the origin image."
replace "origin image" with "original image".

Page 4, line 142 and Equation 5: Error in the upper bound of summation index:
Correction: E_0 =  sum(n=1, n=N, E_{i,i+1}), that is, instead of writing
n = delta for the upper bound, write n = N.

Page 4, Equation 4: How is E_{i,i+1} defined?
By E_{i,i+1} I mean the letter E with "i,i+1" as a subscript.

Page 5, Equation 6: one of the cosines should be a sine. Please, also check
the signs (-,+) of this equation.

Page 5, Equation 9: Replace summation symbol with integration sign.

Pages 7-10, Section 4: Will the software developed for this paper be
 freely available to the scientific community?
 Any plans in making it open source?

Page 11: reference 5: DOI is incorrect.
  Possible typo: "Journal of Astronauties" ==> "Journal of Astronautics"

Reviewer 2 Report

This paper presents a new method of star map synthesis. This method combines star degradation model, image noise model and attitude disturbance model, which can simulate star images with multiple noises. There is no doubt that the method proposed in this paper has certain application prospects in the field of star tracker, but there are some doubts as follows:

1. The method in this paper does not consider the influence of imaging system such as aberration and aperture of optical system. Will these optical parameters affect the authenticity of simulation?

2. This paper only compares the similarity of star map images, but the centroid of stars is the most important in attitude determination. To determine the authenticity of star map simulation, should this paper compare the centroid positioning results of simulated images with the centroid positioning accuracy of real images?

3. The star map taken from the ground is greatly affected by atmospheric refraction, and atmospheric radiation is not corrected in this paper. There are some other studies on this topic, such as:

â‘    Joint Estimation of Stellar Atmospheric Refraction and Star Tracker Attitude, IEEE TRANSACTIONS ON INSTRUMENTATION AND MEASUREMENT, 2022.

â‘¡   Refraction Surface-Based Stellar Atmospheric Refraction Correction and Error Estimation for Terrestrial Star Tracker, IEEE SENSORS JOURNAL, 2022.

The authors are suggested to compare these latest literatures to demonstrate the superiority of the proposed method.

4. In addition to the influence mentioned in the paper, the stellar aberration caused by the earth's revolution and rotation also has influence on the image position of a star. The authors are suggested to analyze the impact of the stellar aberration on the simulation.

5. In this paper, compound Gaussian distribution is used to simulate different kinds of noise in the simulation of star map noise, but there is no description of how to determine the parameters in the noise expression.

6. The Eq. (6) used in this paper is only effective at low speed according to the references: Global field-of-view imaging model and parameter optimization for high dynamic star tracker. Optics Express, 2018. Is the method proposed by the authors effective in high dynamic situations?

Round 2

Reviewer 1 Report

Please see my comments in the attached file.

Reviewer 2 Report

The innovation of the paper needs to be improved.

Author Response

We would like to express gratitude to the reviewer for his careful reading and suggestions.  The impacts of complicated lens effects and other problems will be further studied in our future work.